# Recursive Batch Smoother with Multiple Linearization for One Class of Nonlinear Estimation Problems: Application for Multisensor Navigation Data Fusion

**DOI:** 10.3390/s25247566

**Published:** 2025-12-12

**Authors:** Oleg Stepanov, Alexey Isaev, Elena Dranitsyna, Yulia Litvinenko

**Affiliations:** Faculty of Control Systems and Robotic, ITMO University, 197101 St. Petersburg, Russia; soalax@mail.ru (O.S.); itmo_student@mail.ru (A.I.); ya_litvinenko@mail.ru (Y.L.)

**Keywords:** Bayesian approach, nonrecursive scheme, recursive scheme, filter, smother, extended kalman filter, iterative algorithm, batch algorithm, particle filters, multiple linearization, multisensor navigation data fusion, map-aided navigation

## Abstract

A class of nonlinear filtering problems connected with data fusion from various navigation sensors and a navigation system is considered. A special feature of these problems is that the posterior probability density function (PDF) of the state vector being estimated changes its character from multi-extremal to single-extremal as measurements accumulate. The algorithms based on sequential Monte Carlo methods, which, in principle, provide the possibility of attaining potential accuracy, are computationally complicated, especially when implemented in real time. Traditional recursive algorithms, such as the extended Kalman filter and its iterative modification prove to be inoperable in this case. Two algorithms, devoid of the above drawbacks, are proposed to solve this class of nonlinear filtering problems. The first algorithm, a Recursive Iterative Batch Linearized Smoother (RI-BLS), is essentially a nonrecursive iterative algorithm; at each iteration, it processes all measurements accumulated by the current time of measurement. However, to do this, it uses a recursive procedure: first, the measurements are processed from the first to the current one in the linearized Kalman filter, and then the obtained estimates are processed recursively in reverse time. The second algorithm, a Recursive Iterative Batch Multiple Linearized Smoother (RI-BMLS), is based on the simultaneous use of a set of RI-BLS running in parallel. The application of the proposed algorithms and their advantages are illustrated by a methodological example and solution of the map-aided navigation problem. The calculation of the computational complexity factor has shown that the RI-BLS is more than 15-fold simpler than the particle filter in computational terms, and the RI-BMLS, more than 20-fold with comparable estimation accuracy.

## 1. Introduction

Filtering algorithms designed within the framework of the Bayesian stochastic approach are widely used to solve problems of integrated measurement processing connected with data fusion from various navigation sensors or navigation systems, i.e., multisensor navigation data fusion. They aim to calculate an estimate that is optimal in the mean square sense [1,2,3,4,5,6,7,8,9,10,11]. This estimate is a mathematical expectation corresponding to the posterior probability density function (PDF) for the vector of parameters being estimated. One of the main advantages of such algorithms is the possibility of generating, in addition to the estimate, the corresponding accuracy characteristic in the form of a calculated covariance matrix of estimation errors. When such a matrix is consistent with its real covariance matrix, the filtering algorithm is also considered consistent [5,12].

Usually, algorithms are designed as recursive, which implies the processing of incoming measurements one after another. The problems in which the posterior PDF has a single-extremal form (the function has only one extremum in the domain of the a priori uncertainty, and from here on, such a PDF is referred to as single-extremal) are often solved using recursive Kalman-type algorithms (KTAs) based on the Gaussian approximation of the posterior density [6,11,13,14,15,16]. The simplest of such algorithms are based on the expansion of nonlinear functions, describing the model of a dynamic system and measurements, in a truncated Taylor series. Such algorithms include the linearized [2,6] and extended [2,11] Kalman filters, second-order filters [6,17], polynomial filters [18], third- and higher-order filters [19,20], and their iterative modifications [2,11]. Another approach to the KTA design involves statistical linear regression procedures [21]. They are sometimes called filters that do not require the calculation of derivatives, as when obtaining an approximate description of nonlinear functions using their linear analogs, there is no need to calculate the derivatives [22]. Among such algorithms, also known as sigma-point filters, are, for example, Unscented [23,24], Cubature [25,26], Smart Sampling Kalman Filters [27], and many others [28,29].

Traditional recursive KTAs are ineffective for problems in which the posterior PDF takes a complicated multi-extremal form [30]. To solve them, when designing filtering algorithms, designers use the known recursive relation for the posterior PDF and various methods of its approximation [28]. These algorithms use a significant set of parameters to describe the posterior density. Here, we should mention filters based on the point-mass method and those using poly-Gaussian approximation [31,32,33]. The most widely used are the so-called sequential Monte Carlo methods, also known as particle filters, and their various modifications [34,35,36,37,38]. These algorithms can achieve accuracy close to the potential one, i.e., the accuracy corresponding to the optimal algorithm, but they often prove to be computationally complicated. Various procedures are used to reduce the amount of computation in such algorithms, for example, the importance resampling [39] and the Rao-Blackwellization procedure [40]. Despite this, their computational complexity remains extremely high, which often limits their online application. Thus, the most difficult problems to solve in practice are those in which the posterior PDF has a complicated multi-extremal form. Among various filtering problems connected with multisensor navigation data fusion, we can single out a separate class in which the posterior PDF, being multi-extremal at the initial moments of estimation time, becomes single-extremal. It is this class of problems that is considered in this paper. Such problems arise, for example, during navigation data fusion; among them are the problem of the navigation of a group of autonomous underwater vehicles [30], the beacon navigation problem [41], the map-aided navigation problem [42,43,44,45,46,47], and some others. Traditional recursive algorithms based on the Gaussian approximation of the posterior density turn out to be inefficient for their solution, and recursive algorithms that need a significant set of parameters to describe the posterior density are often computationally intensive, which makes them unsuitable for online applications. In [30,48], we showed that nonrecursive iterative KTAs used to solve such problems can provide the accuracy of optimal estimation and be consistent starting from the moment when the posterior PDF becomes single-extremal. In this case, a batch of all measurements accumulated by the current moment arrives at the input of such algorithms, and this is the reason why they are called batch algorithms [9,30]. Despite all their advantages, these algorithms have two main drawbacks when used to solve the problems under consideration. First, their computational complexity––although often lower than that of algorithms that use a significant set of parameters to describe posterior PDF (for example, particle filters)––remains significant. This is due to the need to invert high-dimensional matrices. Second, the moment when the posterior PDF becomes single-extremal is usually unknown in practice, and its identification is a nontrivial task.

The aim of this paper is to design algorithms that lack the above-mentioned shortcomings. The paper proposes two algorithms.

The first one is economical in a computational sense—with respect to the amount of computations; it is nonrecursive iterative algorithm called the Recursive Iterative Batch Linearized Smoother (RI-BLS). It works due to the fact that instead of inverting a high-dimensional matrix at each iteration, it uses a recursive procedure that ensures the necessary estimates by solving the smoothing problem. That is, the RI-BLS is based on the combined use of recursive and nonrecursive data processing schemes. The RI-BLS has low computational complexity. This algorithm can be used in problems for which the moment when the posterior PDF becomes single-extremal can be determined in advance. At the same time, a challenging problem here is to identify this moment.

The second algorithm is called the Recursive Iterative Batch Multiple Linearized Smoother (RI-BMLS). In some cases, its computational complexity may be somewhat higher than that of the RI-BLS, but the RI-BMLS is capable of determining the moment at which the posterior PDF becomes single-extremal. The proposed RI-BMLS, as well as the RI-BLS, is based on the combined use of recursive and nonrecursive data processing schemes. The essence of the RI-BMLS is the simultaneous use of a set of RI-BLSs, each of which has its own individual linearization point at the initial moment. A set of linearization points, each of which is used in its RI-BLS, is formed in such a way that their environment, all together, should cover the domain of a priori uncertainty. At the moment when the posterior PDF becomes single-extremal, the estimates at the output of the RI-BLS set with various linearization points are grouped into an area corresponding to one extremum, which allows us to identify the moment at which the posterior PDF becomes single-extremal. After that, to save computational resources, the problem is solved with a single recursive iterative Kalman filter. In the case of a single-extremal PDF, it will provide an accuracy close to that of the optimal algorithm. Such algorithms are used to solve the problem of navigation system correction with the use of nonlinear measurements from various sensors, for example, in the navigation of a group of autonomous underwater vehicles, single-beacon navigation, and map-aided navigation.

The paper is structured as follows. In Section 2, the nonlinear filtering problem under study is formulated. Section 3 describes the proposed Recursive Iterative Batch Linearized Smoother, and Section 4 considers the Recursive Iterative Batch Multiple Linearized Smoother. Section 5 formulates a map-aided navigation problem and demonstrates the performance and advantages of the proposed algorithms. Section 6 and Section 7 analyze and summarize the main contributions of this paper.

## 2. Bayesian Statement of the Nonlinear Estimation Problem

Consider the problem of the nonlinear estimation of an *n*-dimensional random vector described by a shaping filter(1)$$x_{k}=f_{k}(x_{k-1})+G_{k}w_{k}+u_{k},$$
using *m*-dimensional vector measurements:(2)$$y_{k}=h_{k}(x_{k})+v_{k}.$$

In these relations, it is assumed that $f_{k}()$ and $h_{k}()$ are known nonlinear *n*- and *m*-dimensional functions describing the dynamics for the state vector and the measurement model; *k* is the discrete time index; $u_{k}=(u_{1,k},\;u_{2,k},\;\ldots \;u_{n,k}{)}^{T}$ is an *n*-dimensional vector of known input signals (for double indexing in subscripts, here and below, the first index denotes the component number, and the second refers to the time); $x_{0}$ is an *n*-dimensional random Gaussian vector with a given PDF $p(x_{0})=N(x_{0};{\bar{x}}_{0},P_{0})$; hereinafter, the notation $N(b;\bar{b},\;B)$ is used for the density of the Gaussian random vector ***b*** with mathematical expectation $\bar{b}$ and covariance matrix ***B***; $w_{k\;}$ is $n_{w}$-dimensional zero-mean discrete Gaussian white noise, independent of $x_{0}$ with a known $n_{w}\times n_{w}$-dimensional covariance matrix $Q_{k}$; $G_{k}$ is an $n\times n_{w}$-dimensional matrix; and $v_{k}$ is *m*-dimensional zero-mean discrete Gaussian white noise, independent of $x_{0}$ and $w_{k}$, with covariance matrix $R_{k}$.

The essence of the estimation problem in the framework of the Bayesian approach is to obtain, in a certain sense, an optimal estimate of state vector ${\hat{x}}_{k}^{opt}(Y_{k}^{c})$ based on measurements $Y_{k}^{c}=(y_{1}^{T},y_{2}^{T},\;\ldots \;,y_{k}^{T}{)}^{T}$ and, if possible, the corresponding conditional covariance matrix of estimation error $P_{k}^{opt}(Y_{k}^{c})$, which characterizes its current accuracy (corresponding to a specific set of measurements $Y_{k}^{c}$).

It is known that optimal, in the mean square sense, estimate ${\hat{x}}_{k}^{opt}(Y_{k}^{c})$ and the corresponding covariance matrix $P(Y_{k}^{c})$ are defined as follows [5]:(3)$${\hat{x}}_{k}^{opt}\left(Y_{k}^{c}\right)=\int x_{k}p\left(x_{k}|Y_{k}^{c}\right)dx_{k},$$(4)$$P(Y_{k}^{c})=\int (x_{k}-{\hat{x}}_{k}^{opt}(Y_{k}^{c}))(x_{k}-{\hat{x}}_{k}^{opt}(Y_{k}^{c}){)}^{T}p(x_{k}|Y_{k}^{c})dx_{k}\;,$$
where $p(x_{k}|Y_{k}^{c})$ is the posterior PDF, conditional to the measurements.

It is clear that in order to find an optimal estimate (3) and covariance matrices (4), we need to have posterior PDF $p(x_{k}|Y_{k}^{c})$, which is difficult to calculate, so that in practice, different methods of its approximation are used to design suboptimal filtering algorithms. The specificity of the nonlinear filtering problem considered in this paper is that the posterior PDF, which has a complicated multi-extremal character at initial moments of estimation time, evolves into a single-extremal form.

Note that this is not uncommon for the Bayesian approach, along with the optimal, in the mean square sense, estimate (3); it is also possible to determine the estimate corresponding to the posterior PDF maximum:(5)$${\hat{x}}_{k}^{opt}\left(Y_{k}^{c}\right)=\underset{x_{k}}{argmax\;}p\left(x_{k}|Y_{k}^{c}\right).$$

It is important to emphasize that in the case where the posterior PDF has one extremum and is close to the Gaussian one, estimates (3) and (5) will be close to each other.

Note that the optimal estimate minimizes the conditional (4) and unconditional covariance matrices of estimation errors defined as [12](6)$$G_{k}^{opt}=E_{p(x_{k},Y_{k})}\left\{\left(x_{k}-{\hat{x}}_{k}^{opt}(Y_{k}^{c})\right){\left(x_{k}-{\hat{x}}_{k}^{opt}(Y_{k}^{c})\right)}^{T}\right\},$$
where $p\left(x_{k},Y_{k}^{c}\right)$ is a joint PDF of vectors $x_{k}$ and $Y_{k}^{c}$, and *E* is the sign of the mathematical expectation with a subscript explaining from which PDF it is calculated.

It was already noted in the Introduction that when designing estimation algorithms, preference is usually given to recursive algorithms based on a processing scheme that is recursive with respect to measurements. Along with recursive algorithms, the desired estimate of the state vector at an arbitrary point in time can be obtained by finding it as part of the vector being estimated, which includes all elements for all *k*. The recursive and nonrecursive algorithms are described in Section A.1 and Section A.2.

Note that the optimal estimate ${\hat{x}}_{k}^{opt}$ does not depend on the processing scheme according to which it is calculated—recursive or nonrecursive. The situation is different when recursive and nonrecursive suboptimal algorithms are used to calculate the estimate. This also applies to algorithms based on linearization, which is the subject of this paper.

The algorithms that are based on the linearized representation of nonlinear functions assume that(7)$$\begin{array}{c} f_{i}(x_{i-1})\approx f_{i}(x_{i}^{lin_{1}})+F_{i}(x_{i-1}-x_{i}^{lin_{1}}),\\ h_{i}(x_{i})\approx h_{i}(x_{i}^{lin_{2}})+H_{i}(x_{i}-x_{i}^{lin_{2}}), \end{array}$$
where i=1,2,…,k, and $F_{i}\;and\;H_{i}$ are Jacobian matrices of functions $f_{i}\left(x_{i-1}\right)\;and\;h_{i}(x_{i})$ with dimensions n×n and m×n calculated at linearization points $x_{i}^{lin_{1}}$ and $x_{i}^{lin_{2}}$.

Suboptimal algorithms based on linearization have two distinctive features. First, the posterior PDF $p(x_{k}|Y_{k}^{c})$ is replaced at each step with its Gaussian approximation described by the estimates generated in the algorithm of estimate ${\hat{x}}_{k}^{SUB}$ and the corresponding covariance $P_{k}^{SUB}$, i.e., $p\left(x_{k}|Y_{k}^{c}\right)\approx N\left(x_{k};{\hat{x}}_{k}^{SUB},P_{k}^{SUB}\right)$. The second feature is determined by the fact that the processing of the current measurement is carried out based on the ideology of designing a linear optimal algorithm [13]. The methods for developing such algorithms with the use of both recursive and nonrecursive schemes are described in Section A.3 and Section A.4.

Recall that in this paper, we consider a class of problems whose specificity lies in the fact that the posterior PDF of the state vector under estimation changes its character, as measurements accumulate, from a multi-extremal to a single-extremal density. The extended Kalman filter (EKF) and iterative extended Kalman filter (IEKF) use a Gaussian approximation of the posterior PDF at each step and involve the estimate from the previous step to calculate the linearization points. It is clear that they are ineffective for the solution of the class of problems under consideration. The reason for this is that at the initial stage (with a small number of accumulated measurements), the form of the posterior PDF differs from the Gaussian one, which leads to significant linearization errors in such well-known algorithms [1,2,3,4,5,6,7,8,9,10] as the EKF and IEKF. In some cases, the accuracy of the EKF and IEKF may be lower than that of the linearized Kalman filter (LKF), whose linearization points at each step *k* are chosen as some fixed values that only depend on the a priori information [30].

As shown in [30,48], in contrast to traditional recursive algorithms, nonrecursive (batch) iterative algorithms based on linearization allow for accuracy close to the potential one, i.e., the accuracy corresponding to the optimal algorithm, in the solution of problems in which the posterior PDF of the estimated state vector changes its character, as measurements accumulate, from multi-extremal to single-extremal.

Note that in the suboptimal nonrecursive (batch) algorithms, the linearization points $X_{k}^{c,lin_{1}}={\left((x_{1}^{lin_{1}}{)}^{T},\;\ldots \;,\;(x_{k}^{lin_{1}}{)}^{T}\right)}^{T}$ depend only on a priori information. As shown in [30], this fact creates prerequisites for increasing the efficiency of the algorithms designed with the use of the nonrecursive scheme, compared to recursive algorithms, when solving problems of the class under consideration.

The simplest batch algorithm, Batch Linearized Smoother (BLS), can be designed if linearization points $x_{i}^{lin_{2}},\;\;i=1,2,\ldots ,k\;$ are chosen as $x_{i}^{lin_{2}}={\bar{x}}_{i}$.

The BLS efficiency can be increased using iterations similar to the way it is achieved in the IEKF. The algorithm designed in this way—Iterative Batch Linearized Smoother (I-BLS)—will correspond to the block diagram shown in Figure 1. The main feature of the I-BLS is that during linearization at fixed points, it repeatedly (iteratively) processes a batch of all measurements accumulated by the current moment of time. In this case, at each iteration, the linearization points are corrected simultaneously taking into account the results obtained at the previous iteration.

As shown in [30], for the class of problems under consideration, the I-BLS, starting from the moment when the PDF becomes single-extremal, is capable of providing accuracy close to the potential one and is consistent. Note that the algorithms called BLS and I-BLS in this paper are called NR-EKF and NR-IEKF in [30]. This is due to the fact that the expressions for the BLS and I-BLS can be obtained by designing the EKF for the solution of the original problems (1) and (2) in its nonrecursive formulation, i.e., at each step *k*, vector $X_{k}^{c}=(x_{0}^{T},\ldots ,x_{k}^{T}{)}^{T}$ is estimated using measurement (2). The main disadvantage of the I-BLS is its high computational complexity.

Further, we present an algorithm called the RI-BLS, based on the combined use of recursive and nonrecursive schemes, which is an economical implementation of the I-BLS in terms of computational complexity.

## 3. Recursive Iterative Batch Linearized Smoother

In the RI-BLS, just as in the case of the I-BLS, after the next *k*-th measurement arrives, all measurements included in the batch of measurements $Y_{k}^{c}=(y_{1}^{T},\;\ldots ,{\;y}_{k}^{T}{)}^{T}$ accumulated by the current moment are processed again (iteratively). Each iteration includes two blocks.

Block 1 implements a recursive solution of the filtering problem for all i=1, 2, …, k using the procedures of the linearized Kalman filter [5], in which linearization points $x_{i}^{lin_{1}}$ and $x_{i}^{lin_{2}}$ are fixed, and in the first iteration (j=1), they depend only on the a priori mathematical expectation ${\bar{x}}_{0}$. In this block, we can highlight a set of calculations for all i=1, 2, …, k:Step 1.Formation of linearization points:$${\bar{x}}_{i}=f_{i}({\bar{x}}_{i-1})+u_{i},\;\;x_{i}^{lin_{1}}={\bar{x}}_{i-1},\;\;x_{i}^{lin_{2}(j)}={\bar{x}}_{i},\;\;j=1;$$
Step 2.Calculation of estimate prediction and its error covariance matrix:
$$\begin{array}{c} {\hat{x}}_{i/i-1}^{\left(j\right)}=f_{i}(x_{i-1}^{lin_{1}})+F_{k}({\hat{x}}_{i-1}^{\left(j\right)}-x_{i-1}^{lin_{1}})+u_{i},\\ \;P_{i/i-1}^{\left(j\right)}=F_{i}^{\left(j\right)}P_{i-1}^{\left(j\right)}(F_{i}^{\left(j\right)}{)}^{T}+G_{i}Q_{i}G_{i}^{T}; \end{array}$$
Step 3.Calculation of measurements prediction and its error covariance matrix:
$$\begin{array}{c} {\hat{y}}_{i}^{\left(j\right)}=h_{i}(x_{i}^{lin_{2}(j)})+H_{i}^{\left(j\right)}({{\hat{x}}^{\left(j\right)}}_{i/i-1}-x_{i}^{lin_{2}(j)}),\\ \;P_{y_{i}}^{\left(j\right)}=H_{i}^{\left(j\right)}P_{i/i-1}^{\left(j\right)}(H_{i}^{\left(j\right)}{)}^{T}+R_{i}; \end{array}$$
Step 4.Calculation of the cross-covariance matrix:
$$P_{x_{i}y_{i}}^{\left(j\right)}=P_{i/i-1}^{\left(j\right)}(H_{i}^{\left(j\right)}{)}^{T};$$
Step 5.Calculation of the gain factor:
$$K_{i}^{\left(j\right)}=P_{x_{i}y_{i}}^{\left(j\right)}P_{y_{i}}^{\left(j\right)};$$
Step 6.Calculation of the estimate and its corresponding covariance matrix:
$$\begin{array}{c} {\hat{x}}_{i}^{\left(j\right)}={{\hat{x}}^{\left(j\right)}}_{i/i-1}+K_{i}^{\left(j\right)}\left(y_{i}-{\hat{y}}_{i}^{\left(j\right)}\right),\\ \;P_{i}^{\left(j\right)}=P_{i/i-1}^{\left(j\right)}-K_{i}^{\left(j\right)}P_{x_{i}y_{i}}^{\left(j\right)}. \end{array}$$

During processing, batches of accumulated measurements $Y_{k}^{c}=(y_{1}^{T},\;\ldots ,\;y_{k}^{T}{)}^{T}$, prediction values ${\hat{x}}_{i/i-1}^{\left(j\right)}$, estimates ${\hat{x}}_{i}^{\left(j\right)}$ calculated in Block 1, and the corresponding covariance matrices $P_{i/i-1}^{\left(j\right)}$ and $P_{i}^{\left(j\right)}$ for each moment of estimation time i=1, 2, ..., k are saved to be used further.

Block 2 implements recursive calculation in reverse order for i=k−1, k−2, ... , 0 of estimates ${\hat{x}}_{i/k}$ corresponding to the solution of the smoothing problem. The second block includes the following calculations:Step 1.Calculation of the smoother transition matrix:$$A_{i}^{\left(j\right)}=P_{i}^{\left(j\right)}(F_{i+1}^{\left(j\right)}{)}^{T}(P_{i+1/i}^{\left(j\right)}{)}^{-1}.$$
Step 2.Calculation of the smoothing estimate and its corresponding covariance matrix:
$$\begin{array}{c} {\hat{x}}_{k/k}^{\left(j\right)}={\hat{x}}_{k}^{\left(j\right)},\;P_{k/k}^{\left(j\right)}=P_{k}^{\left(j\right)},\\ {\hat{x}}_{i/k}^{\left(j\right)}={\hat{x}}_{i}^{\left(j\right)}+A_{i}^{\left(j\right)}({\hat{x}}_{i+1/k}^{\left(j\right)}-{\hat{x}}_{i+1/i}^{\left(j\right)}),\\ P_{i/k}^{\left(j\right)}=P_{i}^{\left(j\right)}+A_{i}^{\left(j\right)}(P_{i+1/k}^{\left(j\right)}-P_{i+1/i}^{\left(j\right)})(A_{i}^{\left(j\right)}{)}^{T}. \end{array}$$

It is important to note that estimates ${\hat{x}}_{i/k}^{\left(1\right)}$ and the corresponding covariance matrices $P_{i/k}^{\left(1\right)}$ obtained at the first iteration of *j* = 1 as a result of the smoothing problem solution coincide with estimates ${\hat{x}}_{i}^{BLS}$ and covariance matrices $P_{i}^{BLS}$, i=0,1, … , k−1 generated in the BLS as part of vector ${\hat{X}}_{k}^{c,BLS}$ and a block of matrix $P_{k}^{c,BLS}$.

Estimates ${\hat{x}}_{i/k}^{\left(j\right)}$ obtained in Block 2, corresponding to the solution of the smoothing problem, are saved and used further during the repeated processing of measurements (in the next iteration) as linearization points $x_{i}^{lin_{2}(j)}$, after which the above procedure is repeated again.

The RI-BLS can be represented as the block diagram shown in Figure 2.

The estimate ${\hat{x}}_{k}^{\left(j\right)}$ and covariance matrix $\;P_{k}^{\left(j\right)}$ obtained at the latest iteration are taken as the estimate and covariance matrix in the RI-BLS at the *k*-th step. The estimate ${\hat{x}}_{k}^{RI-BLS}$ and covariance matrix $P_{k}^{RI-BLS}$ generated in the RI-BLS coincide with the estimate ${\hat{x}}_{k}^{I-BLS}$ and covariance matrix $P_{k}^{I-BLS}$ generated in the I-BLS as part of vector ${\hat{X}}_{k}^{c,I-BLS}$ and a block of matrix $\;P_{k}^{c,I-BLS}$. In addition, when implementing the proposed algorithm, it is not necessary to invert high-dimensional matrices; it is sufficient to invert only the *n* × *n*-dimensional matrices, which allows for a significant reduction in computation. Thus, the RI-BLS can be considered an economical implementation of the I-BLS in terms of computational complexity.

With all the advantages of the I-BLS, the RI-BLS, starting from moment $T^{eff}$, when the posterior PDF takes a single-extremal form, generates an estimate close to the optimal one, in the mean square sense, and satisfies the consistency properties.

The algorithm considered can be used in problems for which the moment when the posterior PDF becomes single-extremal is known in advance. At the same time, in practice, researchers have to deal with problems in which this information is unavailable, particularly in solving navigational problems. The modification of the RI-BLS that allows us to determine the moment $T^{eff}$ is considered below.

## 4. Recursive Iterative Batch Multiple Linearized Smoother

### 4.1. RI-BMLS Description

The main idea of the proposed algorithm is to use a set of RI-BLSs running in parallel, with different linearization points that are selected based on the features of the problem being solved. For example, these points should be selected in such a way that their neighborhoods cumulatively cover the domain of the a priori uncertainty.

It is known [2] that the iterative Kalman filter is aimed to find an estimate corresponding to the maximum of the posterior PDF. The iterative Gauss–Newton procedure used for these purposes does not guarantee global convergence. In the case of a multiextremal posterior distribution, the estimate generated by the iterative algorithm may correspond to a local extremum. However, when the set of RI-BLSs used is sufficiently large and has different linearization points, the estimates they generate will be grouped in areas corresponding to all extrema, both global and local. At estimation time point $T^{eff}$, when the posterior PDF takes the single-extremum form, the RI-BLS estimates will be grouped within a small domain corresponding to a single extremum.

Thus, the process of the RI-BMLS design can be divided into the following steps:
Step 1.Formation of samples $x_{0}^{\left(s\right)},\;s=1,\;2,\;\ldots ,\;S$ from the previously selected domain $\Lambda_{0}$ in which $p(x_{0})$ is different from zero;Parallel start of S
$\mathrm{RI}-\mathrm{BL}S^{\left(s\right)}algorithms,$ for which the linearization points in the first iteration are determined for each according to
$$x_{i}^{\left(s\right)}=f_{i}(x_{i-1}^{\left(s\right)}),\;\;x_{i}^{lin_{1}\left(j\right)}=x_{i-1}^{\left(s\right)},\;\;x_{i}^{lin_{2}\left(j\right)}=x_{i}^{\left(s\right)},\;\;j=1.$$
Step 3.Verification of the following inequalities:
$${\hat{x}}_{\gamma ,k}^{(s)max}-{\hat{x}}_{\gamma ,k}^{(s)min}<D_{\gamma }\;\mathrm{for}\;\gamma =1,2,\;\ldots ,\;n,$$
where ${\hat{x}}_{\gamma ,k}^{(s)max}$ is the maximum value, ${\hat{x}}_{\gamma ,k}^{(s)min}$ is the minimum one among the γ-th components of estimates $x_{k}^{\left(s\right)},\;s=1,2,\;\ldots ,\;S$, and $D_{\gamma }$ denotes the components of a certain predefined vector $D={\left[D_{1},\;D_{2},\;\ldots \;,D_{n}\right]}^{T}$. We will assume the posterior PDF is a single-extremal one if, for each γ=1,2, …, n, the difference between ${\hat{x}}_{\gamma ,k}^{(s)max}$ and ${\hat{x}}_{\gamma ,k}^{(s)min}$ is less than the γ-th component of vector D.

Step 4.If all inequalities are satisfied, then $T^{eff}=t_{k}$

It is clear that the procedure used to identify the extrema of the posterior PDF is approximate, and the correctness of the estimation time point identification will depend on the specific values for the vector D components, which are selected heuristically. However, as will be shown further, the use of such a procedure as part of the RI-BMLS allows for the correct identification of time point $T^{eff}$.

After $T^{eff\;}\;$ has been identified, it becomes possible to continue solving the problem without using the RI-BLS bank and instead use a single recursive IEKF, which, in the case of a single-extremal posterior PDF, will have all the advantages of the RI-BLS and will be simpler in terms of computational complexity.

Thus, the RI-BMLS can be represented as a block diagram shown in Figure 3.

The total time of the problem solution using the proposed RI-BMLS can be divided into two stages.

At the first stage ($t=(0\;,\;T^{eff}]$), when the posterior PDF is multi-extremal, the RI-BLS bank is used. If necessary, it is possible to use the estimate and the estimation error covariance matrix generated in one of the RI-BLSs as the algorithm output at the first stage. However, it should be kept in mind that the algorithm at the first stage is not consistent and the estimation errors can be significant. In this work, we assume that at the first stage, the RI-BMLS solves only the problem of $T^{eff}$ identification and does not generate an estimate and the calculated accuracy characteristic. After $T^{eff}$ has been identified, the estimates and the covariance matrices of all RI-BLSs will coincide and any of them can be used as the initial settings of the IEKF initiated at the next stage.

At the second stage $t=(T^{eff},\infty )$, the problem is solved with the use of the IEKF, and the estimates and estimation error covariance matrices generated in it are the RI-BMLS output. In this case, since the posterior PDF is single-extremal, the estimate generated at the second stage is close in accuracy to the optimal one in the mean square sense, and the algorithm is consistent.

Since the extremum can appear at any point belonging to the domain of a priori uncertainty, the linearization points should be selected in such a way that their neighborhoods cumulatively cover the domain of the a priori uncertainty.

It should be noted that attempts to design a similar algorithm have already been made previously, for example, in [49], but they were not successful. This is primarily due to the fact that the iterative algorithm with multiple linearization was designed with the use of a set of recursive iterative Kalman filters. However, as noted before, the recursive scheme used to solve the class of problems considered here leads to the accumulation of errors caused by linearization. In this paper, we use an RI-BLS—a nonrecursive algorithm—to design an iterative algorithm with multiple linearization.

### 4.2. Methodological Example

Let us explain the essence of the proposed algorithms by considering a simple methodological example.

Assume that it is required to estimate an exponentially correlated sequence $x_{k}$ described by a linear shaping filter:(8)$$x_{k}=Fx_{k-1}+Gw_{k}+u,$$
where $F=e^{-\alpha \Delta t}$, $G=\frac{\sqrt{2\sigma^{2}\alpha }}{\alpha }\left(1-e^{-\alpha \Delta t}\right)$, $\sigma^{2}$ is the sequence variance, α is the value inverse to the correlation interval $\tau_{k}$, and u is a known input signal. Nonlinear measurements (2) have the form(9)$$y_{k}=h_{1}+h_{2}x_{k}+\;h_{3}x_{k}^{2}+\;h_{4}x_{k}^{3}+v_{k},$$
where $h_{1},\;h_{2},\;h_{3},\;and\;h_{4}$ are known coefficients.

The simulation was carried out with the following parameters: σ=1.5; $\alpha =0.1\;s^{-1};$ r=0.1, where $\;r^{2}$ is the measurement variance; $h_{1}=0.0875,$ $h_{2}=-0.1825,$ $h_{3}=0.01,$ $h_{4}=0.01;$ u=1; sampling interval Δt=1 s; and simulation time T=10 s.

The problem was simulated and solved for one sample (run) with a set of S=46 $\mathrm{RI}-\mathrm{BL}S^{\left(s\right)},$ where $s=\bar{1..S}$. The linearization points at the initial moment of estimation time were uniformly distributed over the domain $\Lambda^{0}=\left[-3\sigma ;3\sigma \right]$ with a step of $\theta^{lin}=0.2$.

The results of the RI-BMLS operation in the form of the true values of the estimated sequence and RI-BLS estimates generated for a different number of processed measurements are presented in Figure 4.

In Figure 4, the blue solid lines represent the graphs of estimates ${\hat{x}}_{k}^{\left(s\right)}$ obtained with the $\mathrm{RI}-\mathrm{BL}S^{\left(s\right)}$ output. Note that at the initial moment, linearization points were taken as estimates, i.e., ${\hat{x}}_{0}^{\left(s\right)}=x_{0}^{\left(s\right)}$. The red solid line shows the true values of the estimated sequence, and the dotted purple line indicates the identified point of estimation time $T^{eff}$.

The graph in Figure 4 shows that the estimates were initially grouped within two areas, but as measurements accumulated, these areas began to converge with each other and with the true value of the sequence being estimated. At a certain point in time, when *k* = 7, the estimates at the $\mathrm{RI}-\mathrm{BL}S^{\left(s\right)}$ output clustered within a small domain corresponding to the true value of the estimated variable, allowing us to correctly identify $T^{eff}$. The posterior PDF then became single-extremal. Figure 5 shows the graphs of the posterior PDF obtained with the use of sequential Monte Carlo methods [34] at k=1, 3, 5, 7.

It is evident from Figure 4 that after processing the measurements at step *k* = 7, the posterior PDF becomes single-extremal, as shown in Figure 5d. At the same time, it is evident from Figure 4 that at the same time moment, the estimates at the output of the whole set of $\mathrm{RI}-\mathrm{BL}S^{\left(s\right)}$ are grouped within a small area corresponding to the real value of the estimated quantity, which made it possible to correctly identify the moment of time $T^{eff}$.

## 5. Map-Aided Navigation

Let us now consider one of the possible options for the practical application of the proposed algorithm to map-aided navigation [42,43,44,45,46,47,49]. This problem is an example of the data fusion from a navigation system and a sensor of the Earth gravity field. Following [42], the problem of map-aided navigation is formulated with a Bayesian framework as a nonlinear filtering problem. It is assumed that there is a navigation system (NS) that generates vehicle coordinates $y_{k}^{NS}={\left[\begin{array}{cc} y_{k}^{NS(1)}, & y_{k}^{NS(2)} \end{array}\right]}^{T}$ in the following form:(10)$$y_{k}^{NS}=x_{k}^{O}+\Delta_{k},$$
where $x_{k}^{O}=[x_{1,k}^{O},\;x_{2,k}^{O}{]}^{T}$ are unknown vehicle coordinates, and $\Delta_{k}=[\Delta_{1,k},\;\Delta_{2,k}{]}^{T}$ are the NS errors.

In addition, we assume that there is a sensor of some geophysical field and a corresponding digital map.

Thus, we can write the measurements as(11)$$y_{k}=\varphi (x_{k}^{O})+v_{k},$$
where function $\varphi (x_{k}^{O})$ determines the dependence of the geophysical field on the vehicle coordinates, and $v_{k}$ are the measurement errors.

By replacing $x_{k}^{O}$ with $\left(y_{k}^{NS}-\Delta_{k}\right)$, we can formulate the following problem: estimate $\Delta_{k}$ using measurements(12)$$y_{k}=\varphi \left(y_{k}^{NS}-\Delta_{k}\right)+v_{k}.$$

Consider the simplest case: when the NS errors for each coordinate are described by a random zero-mean Gaussian value with variances $(\sigma^{\Delta }{)}^{2}$, and the measurement errors are described by Gaussian white noise with variances $r^{2}$. In this case, we need to estimate the two-dimensional vector(13)$$x_{k}=\Delta_{k}=\Delta_{k-1}=\Delta$$
with measurements(14)$$y_{k}=\varphi (y_{k}^{NS}-\Delta )+v_{k}=h_{k}\left(\Delta \right)+v_{k}.$$

We also suppose that vector $x_{k}$ and errors $v_{k}$ are independent of each other. Under the assumptions made, we can write the Formula (A7) as(15)$$J\left(\Delta \right)=\frac{(\Delta_{1,k}{)}^{2}+(\Delta_{2,k}{)}^{2}}{(\sigma^{\Delta }{)}^{2}}+\sum_{i=1}^{k}\frac{{\left(y_{i}-h_{i}(\Delta_{1,k},\Delta_{2,k})\right)}^{2}}{r^{2}}$$

The simulation was performed for a gravity anomaly field generated using the EGM 2008 model [50]. In this case, $\varphi (x_{k}^{O})$ describes the dependence of gravity anomalies on the coordinates. The isolines of gravity anomalies are shown in mGal in Figure 6.

The path was assumed to be fixed, and the following parameters were used in the simulation: $\sigma^{\Delta }=1$ km; r=0.5 mGal; the distance between measurements was δ=300 m; and the path length was 24 km.

First, we demonstrate the solution of the problem under consideration using the proposed algorithm with multiple linearization for one sample. To design the algorithm, we set the linearization points at the initial moment at the nodes of a uniform grid on the region of a priori uncertainty. Figure 7 shows the isolines of the a priori PDF, and the red dots indicate the position of the linearization points.

Figure 8 shows the isolines of the posterior PDFs at the estimation time moments k=1, k=15, and k=30; the red dots indicate the estimates generated by the $\mathrm{RI}-\mathrm{BL}S^{\left(s\right)}.$

As can be seen from the simulation results, the posterior PDF, which has, at the initial estimation time moments, a multi-extremal form significantly different from the Gaussian one, becomes single-extremal as measurements accumulate, and the estimates converge within the neighborhood of a small domain, which makes it possible to identify $T^{eff}$.

Note that the RI-BMLS, representing a set of S=441 nonrecursive RI-BLS, was used only methodologically to clarify its essence. The total complexity of the designed algorithm with multiple linearization turns out to be extremely high, which does not allow it to be applied in online mode. In this connection, to solve the problem under consideration, we designed the RI-BMLS with only S=9 linearization points or nine parallel $\mathrm{RI}-\mathrm{BL}S^{\left(s\right)},$ which is the same. Figure 9 shows the isolines of the a priori PDF graph; the red dots represent the positions of the linearization points at the initial moment of time.

Using statistical testing and predictive simulation based on L=500 samples, consider the simulation results for the following algorithms: the RI-BLS and RI-BMLS proposed in this paper, two traditional recursive linearization-based algorithms—EKF and IEKF—and an algorithm based on sequential Monte Carlo methods—particle filter (PF). The PF aims to calculate the optimal estimate, in the mean square sense. For each μ-th algorithm, according to the methodology in [12], we obtained the real $G_{k}^{\mu }$ and calculated covariance matrices ${\tilde{G}}_{k}^{\mu }$:(16)$$G_{k}^{\mu }\approx \frac{1}{L}\sum_{j=1}^{L}(x_{k}^{\left(j\right)}-{\hat{x}}_{k}^{\mu }(Y_{k}^{c,(j)}))(x_{k}^{\left(j\right)}-{\hat{x}}_{k}^{\mu }\left(Y_{k}^{c,\left(j\right)}\right){)}^{T},$$(17)$${\tilde{G}}_{k}^{\mu }\approx \frac{1}{L}\sum_{j=1}^{L}P_{k}^{\mu }(Y_{k}^{c,(j)}),$$
where $x_{k}^{\left(j\right)}$ and $Y_{k}^{c,(j)}$, $j=\bar{1.L}$, are the samples of random vectors obtained by simulation according to (12) and (13), and ${\hat{x}}_{k}^{\mu }(Y_{k}^{c,(j)})$ and $P_{k}^{\mu }(Y_{k}^{c,(j)})$ are the estimates and calculated covariance matrices obtained for the *j*-th realizations of $x_{k}^{\left(j\right)}$ and $Y_{k}^{c,(j)}$, $j=\bar{1.L}$.

Using (16) and (17), we calculated the real $\sqrt{G_{k}^{R,\mu }}$ and calculated $\sqrt{{\tilde{G}}_{k}^{R,\mu }}$ radial errors of the estimate for each of the coordinates:(18)$$\sqrt{G_{k}^{R,\mu }}=\sqrt{G_{k}^{x_{1},\mu }+G_{k}^{x_{2},\mu }},\;\sqrt{{\tilde{G}}_{k}^{R,\mu }}=\sqrt{{\tilde{G}}_{k}^{x_{1},\mu }+{\tilde{G}}_{k}^{x_{2},\mu }},$$
where $G_{k}^{x_{l},\mu }\;$ and $\;{\tilde{G}}_{k}^{x_{l},\mu },\;l=1,2$, are the real and calculated variances of estimation errors ${\hat{x}}_{1,k}$ and ${\hat{x}}_{2,k}$ averaged over L samples.

Also, for the purpose of comparing the computational complexity, we calculated the corresponding coefficient:(19)$$T^{\mu }=\frac{\tau^{\mu }-\tau^{*}}{\tau^{*}},$$
where $\tau^{\mu }=\frac{1}{L}\sum_{j=1}^{L}t_{j}^{\mu },\;\tau^{*}=\frac{1}{L}\sum_{j=1}^{L}t_{j}^{*}$, $t_{j}^{\mu }$ is the time spent by the computer to solve the estimation problem using the analyzed algorithm, and $t_{j}^{*}$ is the time corresponding to the EKF algorithm, which requires the minimum time of all the compared algorithms.

The formula below was used to calculate the value of(20)$${\bar{T}}^{eff}=\frac{1}{L}\sum_{j=1}^{L}T^{eff(j)},$$
which characterizes the time averaged over a set of samples, starting from which the posterior PDF becomes single-extremal.

In Figure 10, the solid line shows the calculation results for the real radial errors, and the dotted line, the calculated radial errors. Blue color corresponds to the EKF (no. 1); green—the IEKF (no. 2); black—RI-BLS (no. 3); purple –RI-BMLS (no. 4); and red—PF (no. 5). The purple dotted line corresponds to ${\bar{T}}^{eff}$. The real radial errors for the EKF and IEKF are the largest compared to other methods; moreover, starting from the moment when the posterior PDF becomes single-extremal, they begin to grow indefinitely. On the contrary, algorithms RI-BLS and RI-BMLS demonstrate high accuracy; in doing so, the actual radial error coincides with the calculated one after ${\bar{T}}^{eff}$ is reached.

The simulation results show that ${\bar{T}}^{eff}$ almost coincides with the moment when the RI-BLS reaches the PF accuracy and becomes consistent, which indicates that, starting from this moment, the posterior PDF becomes single-extremal. At the same time, the recursive EKF and IEKF turned out to be inefficient in solving map-aided navigation problems

Table 1 presents the values of the computational complexity factor calculated using (20) for the compared algorithms.

The calculation of the computational complexity factor has shown that the RI-BLS is 15-fold simpler in computational terms than the PF. Note that the RI-BMLS using a set of $\mathrm{RI}-\mathrm{BL}S^{\left(s\right)}$ turned out to be simpler than the RI-BLS by a factor of 1.5 and more than 20-fold simpler than the PF.

## 6. Discussion

It should be noted that the algorithms described in this paper are universal since they are intended to solve a wide class of problems in which the posterior PDF takes a single-extremal form as measurements accumulate. This class includes, among others, problems associated with the navigation data fusion, for example, the problem of map-aided navigation discussed above, the group navigation problem of autonomous underwater vehicles described in [30], the single-beacon navigation problem [51], and others. We would like to emphasize that the idea of using multiple linearization in designing filtering algorithms, which underlies RI-BMLS, is not new in itself. Such algorithms have already been proposed earlier, for example, in [41,49]. But unlike the RI-BMLS proposed in this paper, they were designed with the use of a recursive scheme, which limits their application to the class of problems discussed in this paper.

It is clear that the comparison results of the computational complexities presented in Table 1 are due to the specific features of the map-aided navigation problem being solved, namely, the fact that the type of posterior PDF becomes single-extremal in a rather short period of time. But if this is not the case and the posterior PDF remains multi-extremal for a long time, the computational complexity of the RI-BMLS may be high due to the nonrecursive character of the $\mathrm{RI}-\mathrm{BL}S^{\left(s\right)}$.

Another possible way to reduce the computational complexity of the RI-BLS and, as a consequence, RI-BMLS is to use incremental smoothing procedures developed with the factor-graph optimization method [52], particularly in the iSAM2 algorithm [53].

In principle, the proposed RI-BMLS can be constructed without using RI-BLS, for example, with the simultaneous use of a set of parallel iSAM2 algorithms or some other computationally efficient nonrecursive algorithms, but this issue is beyond the scope of this paper.

One of the RI-BMLS disadvantages is its low accuracy and the fact that it does not satisfy the consistency properties before the moment of time $T^{eff}$. Yet, this disadvantage can also be overcome if the posterior PDF in it is described on the time interval $t=(0\;,\;T^{eff}]$ as a sum of Gaussian densities determined by the estimates and covariance matrices generated in the RI-BLS bank. However, the computational complexity of the RI-BMLS in this case can increase significantly. Indeed, the accumulation of measurements makes the algorithm implementation more complicated. However, the algorithms discussed are intended to be used in a special class of problems in which a batch of measurements is only necessary at the initial stage of the solution. At the same time, it should be noted that it is the accumulation of data that makes the algorithm efficient.

Moreover, the computational complexity of the algorithm increases with the increase in the dimension of the state vector. This is especially true for solutions of nonlinear problems. However, the RI-BMLS algorithm developed allows us to limit the growth of computational complexity compared to, for example, a particle filter.

## 7. Conclusions

A class of nonlinear filtering problems connected with data fusion from various navigation sensors and a navigation system has been considered. A special feature of these problems is that the posterior PDF of the state vector being estimated changes its character from multi-extremal to single-extremal as measurements accumulate.

Algorithms based on sequential Monte Carlo methods (particle filter), which in principle provide the possibility of attaining potential accuracy, corresponding to the optimal accuracy in the mean square sense of the estimate, are computationally complicated, especially when implemented in real time. Traditional recursive algorithms, such as the extended Kalman filter and its iterative modification prove to be inoperable in this case.

Two algorithms, devoid of the above drawbacks, are proposed to solve this class of nonlinear filtering problems: the RI-BLS and the RI-BMLS.

The first algorithm, RI-BLS, is essentially a nonrecursive iterative algorithm; at each iteration, it processes all measurements accumulated by the current time of measurement. However, to achieve this, it uses a recursive procedure: first, the measurements are processed from the first to the current one in the linearized Kalman filter, and then the obtained estimates are processed recursively in reverse time. The RI-BLS, like the I-BLS, provides accuracy close to the potential one and has the consistency property starting from the moment of time $T^{eff}$, when the posterior PDF becomes single-extremal. Along with this, when implementing the RI-BLS, it is not necessary to invert high-dimensional matrices *nk × nk*, but it is sufficient to invert only n-dimensional matrices, which allows us to significantly reduce the amount of calculation compared to the I-BLS. This algorithm can be used in problems for which the moment when the posterior PDF becomes single-extremal can be determined in advance. In practice, this is not always the case.

The second algorithm is free from this drawback. It is the RI-BMLS based on the RI-BLS that makes it possible to determine the moment $T^{eff}$. The essence of the RI-BMLS lies in the simultaneous use of a set of RI-BLSs running in parallel; the linearization points for each RI-BLS are selected based on the peculiarities of the problem being solved. The estimates generated by a set of RI-BLSs are grouped in the domains corresponding to the PDF extrema, both local and global. At moment of time $T^{eff}$, when the posterior PDF becomes single-extremal, the estimates at the RI-BLS output are clustered within a small domain corresponding to a single extremum. This allows us to identify $T^{eff}$, after which the problem is solved using one IEKF, which, in the case of a single-extremal PDF, is consistent and provides accuracy close to the potential one, corresponding to the optimal algorithm.

The application of the proposed algorithms is illustrated with a methodological example and the solution of the map-aided navigation problem. The effectiveness of the proposed algorithms is demonstrated by solving the problem of data fusion from a navigation system and a sensor of the Earth gravity field, the digital map of which is available aboard the vessel. The calculation of the computational complexity factor showed that the RI-BLS is more than 15-fold simpler than the particle filter in computational terms, and the RI-BMLS is more than 20-fold with a comparable estimation accuracy.

## Figures and Tables

**Figure 1 sensors-25-07566-f001:**
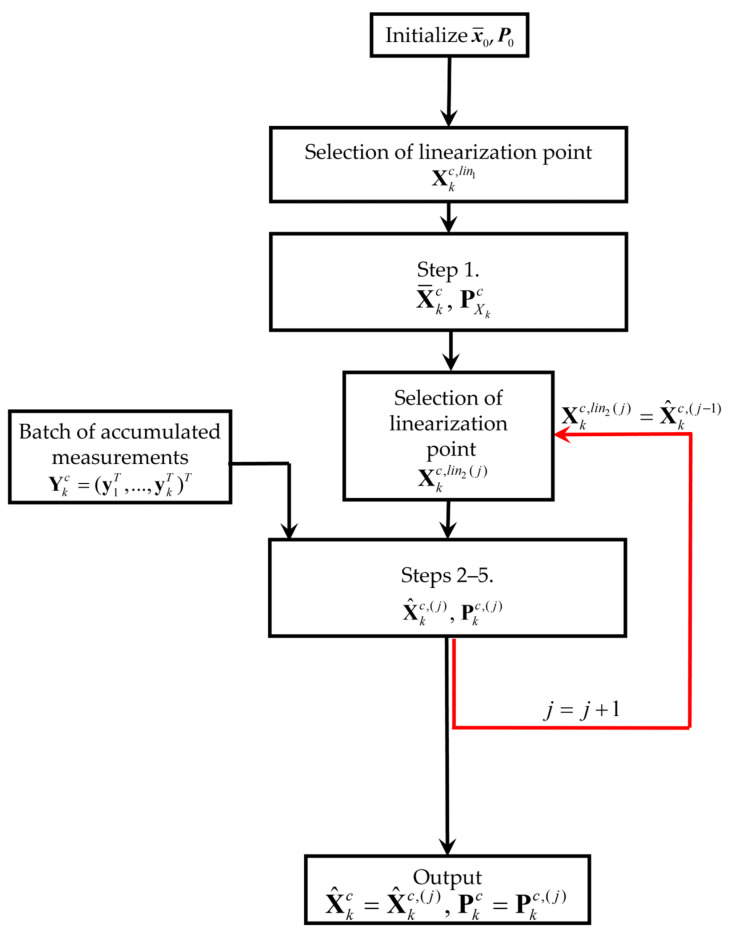
Block diagram of the Iterative Batch Linearized Smoother.

**Figure 2 sensors-25-07566-f002:**
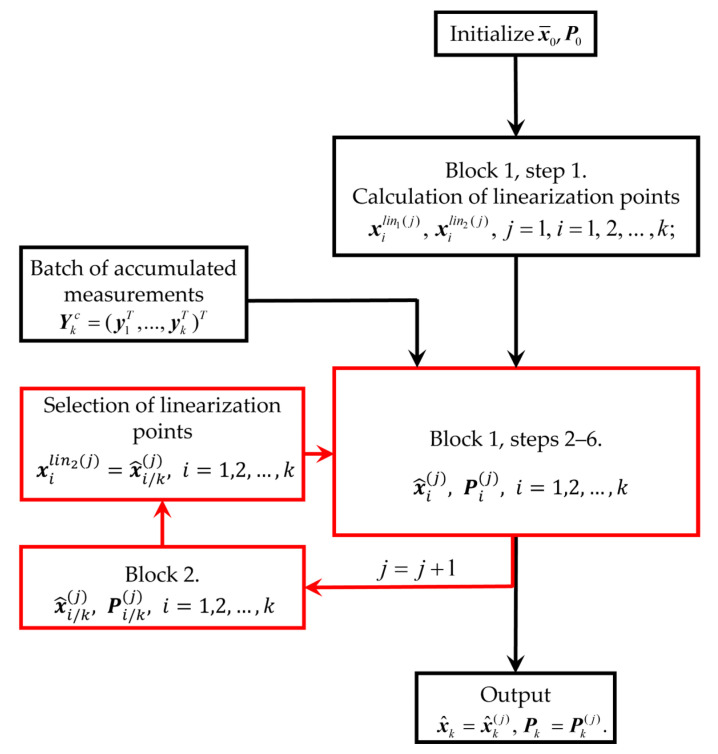
Block diagram of RI-BLS.

**Figure 3 sensors-25-07566-f003:**
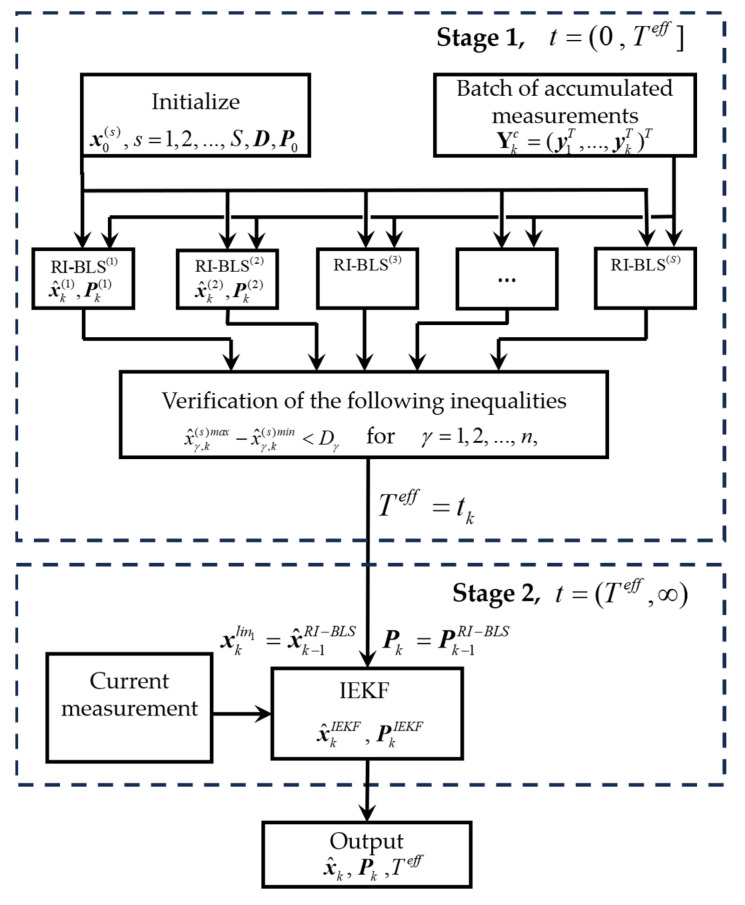
Block diagram of the RI-BMLS.

**Figure 4 sensors-25-07566-f004:**
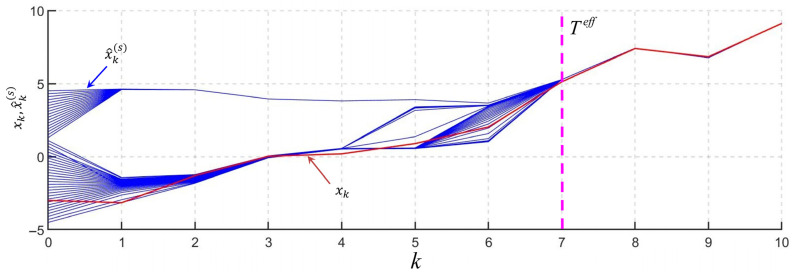
RI-BMLS estimates for different numbers of measurements.

**Figure 5 sensors-25-07566-f005:**
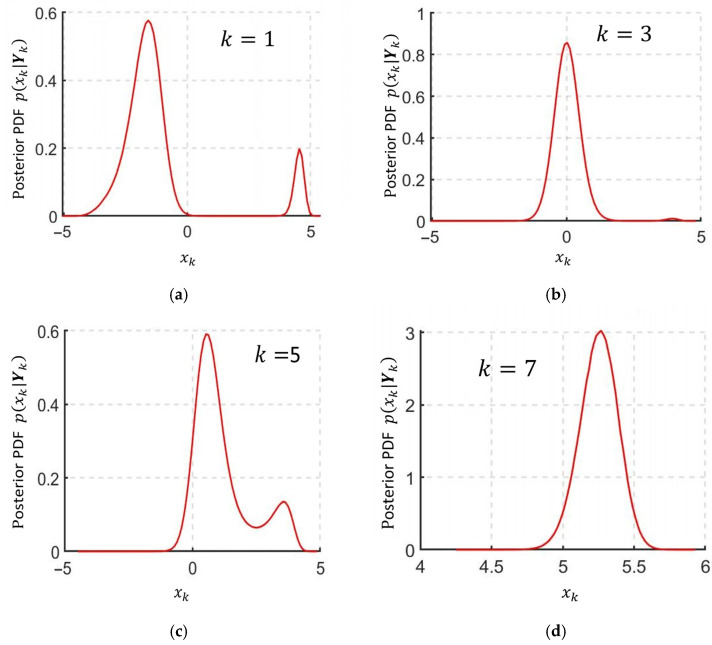
Graphs of posterior PDFs for different estimation time moments: (**a**) *k* = 1; (**b**) *k* = 3; (**c**) *k* = 5; (**d**) *k* = 7.

**Figure 6 sensors-25-07566-f006:**
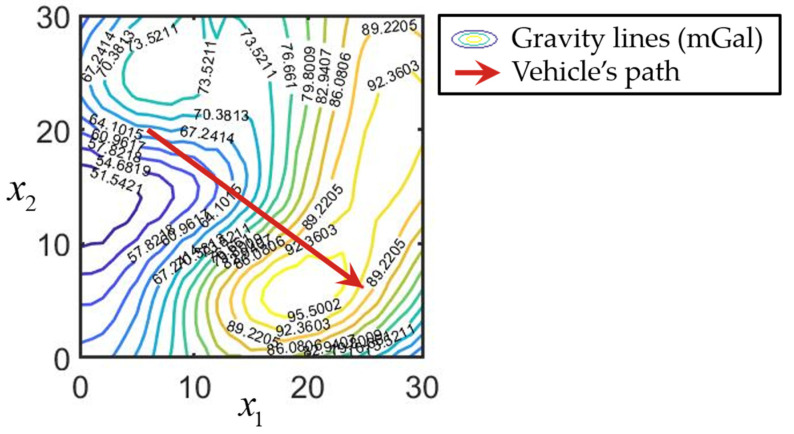
The scheme of the isolines of the gravity anomalies for the region and the vehicle’s path.

**Figure 7 sensors-25-07566-f007:**
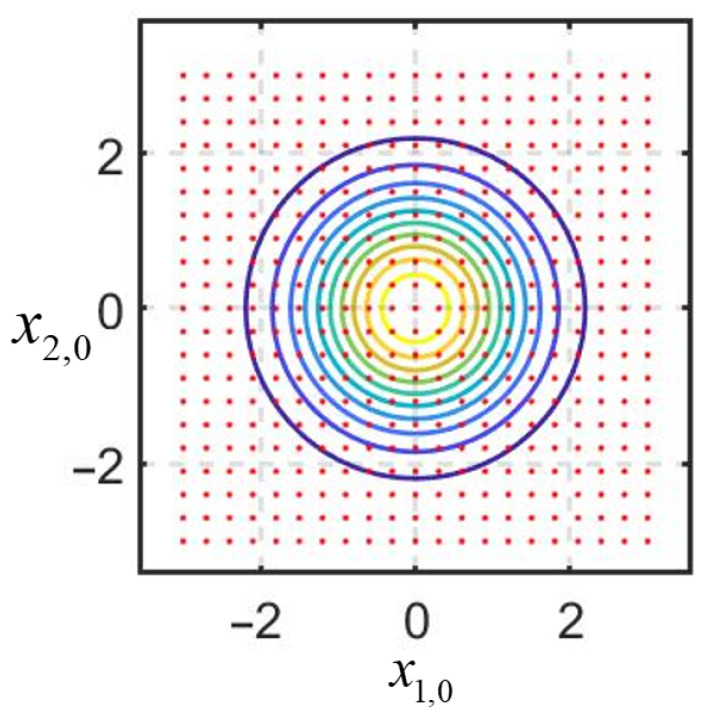
Position of the linearization points in the a priori uncertainty domain.

**Figure 8 sensors-25-07566-f008:**
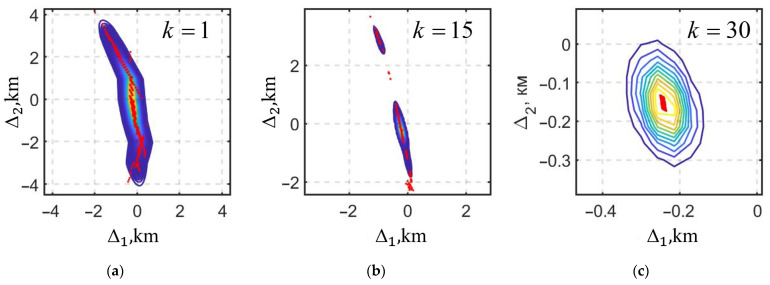
The isolines of the posterior PDFs at different estimation time moments: (**a**) k = 1; (**b**) k = 15; (**c**) k = 30.

**Figure 9 sensors-25-07566-f009:**
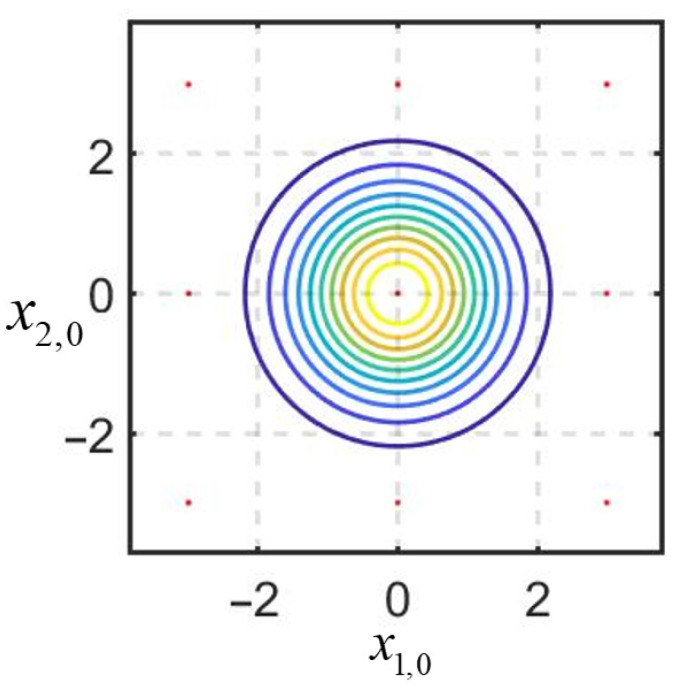
Position of the linearization points in the a priori uncertainty domain.

**Figure 10 sensors-25-07566-f010:**
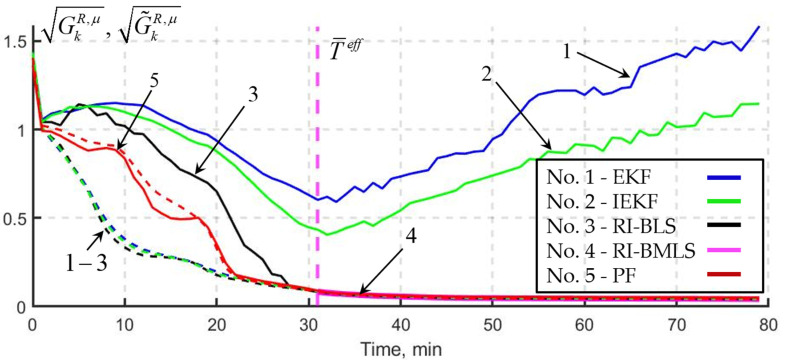
Calculation results for the real and calculated radial errors.

**Table 1 sensors-25-07566-t001:** The values of the computational complexity factor of the algorithms.

Algorithm	Computational Complexity Factor
EKF	0
IEKF	3.2
RI-BLS	181
RI-BMLS	121
PF	2496

## Data Availability

The original contributions presented in this study are included in the article. Further inquiries can be directed to the corresponding author.

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
