# Peer review of "Recursive Batch Smoother with Multiple Linearization for One Class of Nonlinear Estimation Problems: Application for Multisensor Navigation Data Fusion"

_sensors, 2025, doi:10.3390/s25247566_

Round 1

Reviewer 1 Report

Comments and Suggestions for Authors

The paper focuses on nonlinear estimation problems within the Bayesian approach. The specific challenge is a class of problems where the posterior probability density function (PDF)—which represents the likelihood of different states—has a "complicated multi-extremal form" (many possible peaks or solutions) at the beginning, but "becomes single-extremal" (converges to one clear solution) as more measurements are collected over time. The authors propose 2 algorithms Recursive Iterative Batch Linearized Smoother (RI-BLS), which is should be computationally economical; the : Recursive Iterative Batch Multiple Linearized Smoother (RI-BMLS) which is an improvement of the RI-BLS. The algorithms are applied to a map-aided navigation problem.

The paper is generally well presented. I guess the authors are here developping the theory and future publications will be focusing on the application.

There are no major issues. As minor comments, I would propose that the paper could be simplified by putting more common knowledge in the appendices, sections 2 and 3 could be easier to read.Figure 5 is important and need more explanations in the caption. Same for Figure 7 and Figure 11. Conclusions: Try to emphasize the main results to make it more relevant for the reader to go through all the maths. Abstract: You need to shorten it and use the main results underline in the conclusions to "showcase" your work. 

Author Response

We sincerely thank the reviewer for the careful reading of our manuscript and for the constructive comments and suggestions. We hope that the feedback has helped us to improve the clarity, precision, and scientific rigor of the paper.

All changes in the text are highlighted in yellow.

Comment 1: As minor comments, I would propose that the paper could be simplified by putting more common knowledge in the appendices, sections 2 and 3 could be easier to read.

Response 1: We thank the reviewer for this comment. We have, accordingly, revised the manuscript. Descriptions of optimal recursive, nonrecursive, suboptimal recursive and nonrecursive (batch) algorithms based on linearization have moved to Appendix A.

Comment 2: Figure 5 is important and need more explanations in the caption. Same for Figure 7 and Figure 11.

Response 2: We thank the reviewer for this comment

We have added the explanation for Figure 5 (now Figure 4), see lines 384-389:

The graph in Figure 4 shows that the estimates were initially grouped within two areas, but as measurements accumulated, these areas began to converge with each other and with the true value of the sequence being estimated. At a certain point in time, when k = 7, the estimates at the  output clustered within a small domain corresponding to the true value of the estimated variable, allowing us to correctly identify . The posterior PDF then became single-extremal.

We have added the explanation for Figure 7 (now Figure 6), see lines 421-426:

The simulation was performed with a gravity field mapped using the EGM 2008 model [50]. The isolines of gravity anomalies are shown in mGal in Figure 6. The path was assumed to be fixed, and the following parameters were used in the simulation:  km;  mGal; the distance between measurements was ; and the path length was 24 km.

We have added the explanation for Figure 11 (now Figure 10), see lines 478-482:

The real radial errors for the EKF and IEKF are the largest as compared to other methods; moreover, starting from the moment when the posterior PDF becomes single-extremal, they begin to grow indefinitely. On the contrary, the algorithms  and  demonstrate high accuracy; in so doing, the actual radial error coincides with the calculated one after  is reached.

Comment 3: Conclusions: Try to emphasize the main results to make it more relevant for the reader to go through all the maths.

Response 3: We thank the reviewer for this comment. The text of the conclusion has been revised as follows:

A class of nonlinear filtering problems connected with data fusion from various navigation sensors and a navigation system has been considered. A special feature of these problems is that the posterior PDF of the state vector being estimated changes its character from multi-extremal to single-extremal as measurements accumulate.

Algorithms based on sequential Monte Carlo methods, which, in principle, provide the possibility of attaining potential accuracy, corresponding to the optimal accuracy in the mean square sense of the estimate are computationally complicated, especially when implemented in real time. Traditional recursive algorithms, such as the Extended Kalman Filter and its iterative modification prove to be inoperable in this case.

Two algorithms, devoid of the above drawbacks, are proposed to solve this class of nonlinear filtering problems: the RI-BLS and the RI-BMLS.

The first algorithm, RI-BLS, is essentially a nonrecursive iterative algorithm; at each iteration, it processes all measurements accumulated by the current time of measurement. However, to do this, it uses a recursive procedure: first, the measurements are processed from the first to the current one in the linearized Kalman filter, and then the obtained estimates are processed recursively in reverse time. The RI-BLS, like the I-BLS, provides accuracy close to the potential one and has the consistency property starting from the moment of time , when the posterior PDF becomes single-extremal. Along with this, when implementing the RI-BLS, it is not necessary to invert high-dimensional matrices nk × nk, but it is sufficient to invert only n-dimensional matrices, which allows us to significantly reduce the amount of calculations compared to the I-BLS. This algorithm can be used in problems for which the moment when the posterior PDF becomes single-extremal can be determined in advance. In practice, this is not always the case.

The second algorithm is free from this drawback. It is the RI-BMLS based on the RI-BLS that makes it possible to determine the moment . The essence of the RI-BMLS lies in the simultaneous use of a set of RI-BLS running in parallel; the linearization points for each RI-BLS are selected based on the peculiarities of the problem being solved. The estimates generated by a set of RI-BLS are grouped in the domains corresponding to the PDF extrema, both local and global. At the moment of time , when the posterior PDF becomes single-extremal, the estimates at the RI-BLS output are clustered within a small domain corresponding to a single extremum. This allows identifying , after which the problem is solved using one IEKF, which, in the case of a single-extremal PDF, is consistent and provides accuracy close to the potential one, corresponding to the optimal algorithm.

The application of the proposed algorithms has been illustrated by a methodological example and the solution of the map-aided navigation problem. The effectiveness of the proposed algorithms is demonstrated by solving the problem of data fusion from a navigation system and a sensor of the Earth gravity field, the digital map of which is available on board the vessel. Calculation of the computational complexity factor has shown that the RI-BLS is more than 15-fold simpler than the particle filter in computational terms, and the RI-BMLS, more than 20-fold with comparable estimation accuracy.

Comment 4: Abstract: You need to shorten it and use the main results underline in the conclusions to "showcase" your work. 

Response 3: We thank the reviewer for this comment. The text of the abstract has been revised as follows:

A class of nonlinear filtering problems connected with data fusion from various navigation sensors and a navigation system is considered. A special feature of these problems is that the posterior probability density function (PDF) of the state vector being estimated changes its character from multi-extremal to single-extremal as measurements accumulate. The algorithms based on sequential Monte Carlo methods, which, in principle, provide the possibility of attaining potential accuracy, are computationally complicated, especially when implemented in real time. Traditional recursive algorithms, such as the Extended Kalman Filter and its iterative modification prove to be inoperable in this case. Two algorithms, devoid of the above drawbacks, are proposed to solve this class of nonlinear filtering problems. The first algorithm, a Recursive Iterative Batch Linearized Smoother (RI-BLS), is essentially a nonrecursive iterative algorithm; at each iteration, it processes all measurements accumulated by the current time of measurement. However, to do this, it uses a recursive procedure: first, the measurements are processed from the first to the current one in the linearized Kalman filter, and then the obtained estimates are processed recursively in reverse time. The second algorithm, a Recursive Iterative Batch Multiple Linearized Smoother (RI-BMLS), is based on the simultaneous use of a set of RI-BLS running in parallel. The application of the proposed algorithms and their advantages are illustrated by a methodological example and the solution of the map-aided navigation problem. Calculation of the computational complexity factor has shown that the RI-BLS is more than 15-fold simpler than the particle filter in computational terms, and the RI-BMLS, more than 20-fold with comparable estimation accuracy.

Reviewer 2 Report

Comments and Suggestions for Authors

The authors focus on the special nonlinear estimation problem where the PDF gradually evolves from multi-modal to unimodal, filling the efficiency gap of traditional Kalman filter-based algorithms in such scenarios, and is suitable for practical engineering needs such as navigation data processing. This manuscript still has many inadequacies. I hope the following comments can help to improve the manuscript.

  1. Is it more appropriate to use p(xk|Yk) rather than p(xk/Yk) to represent conditional probability?
  2. In the RI-BMLS algorithm, how to ensure that the selected linearization points cover the regions where the extrema of the probability density occur?
  3. When the parameters to be estimated are three-dimensional or higher-dimensional vectors, will the number of linearization points selected by the RI-BMLS algorithm increase exponentially, and will the computational efficiency and performance decrease?
  4. Both the RI-BLS and RI-BMLS algorithms require a batch of accumulated measurement data. Does this limit their application in real-time data processing?

Author Response

Dear Reviewer,

Thank you very much for giving us the opportunity to revise our manuscript. We sincerely appreciate the time and effort that you have dedicated to providing insightful feedback on our work. We have carefully considered all the comments and found them highly constructive and helpful for us to improve the quality of our paper.

All changes in the text are highlighted in yellow.

Comment 1: Is it more appropriate to use p(xk|Yk) rather than p(xk/Yk) to represent conditional probability?

Response 1: Thanks to the reviewer for this important remark. We replaced p(xk/Yk) with p(xk|Yk) in the text.

Comment 2: In the RI-BMLS algorithm, how to ensure that the selected linearization points cover the regions where the extrema of the probability density occur?

Response 2: We thank the reviewer for this essential remark. The relevant explanations are given in the text, see lines 351-353:

Since the extremum can appear at any point belonging to the domain of a priori uncertainty, the linearization points should be selected in such a way that their neighborhoods cumulatively cover the domain of the a priori uncertainty.

Comment 3. When the parameters to be estimated are three-dimensional or higher-dimensional vectors, will the number of linearization points selected by the RI-BMLS algorithm increase exponentially, and will the computational efficiency and performance decrease?

Response 3: We thank the reviewer for this insightful comment. Really, the computational complexity of the algorithm increases with the increase in the state vector dimension. This is especially true for solution of nonlinear problems. However, the RI-BMLS algorithm developed allows us to limit the growth of computational complexity compared to, for example, a particle filter. The relevant explanations are given in the text, lines 546-549.

Comment 4: Both the RI-BLS and RI-BMLS algorithms require a batch of accumulated measurement data. Does this limit their application in real-time data processing?

Response 4: We thank the reviewer for this essential remark. Indeed, accumulation of measurements makes the algorithm implementation more complicated. However, the algorithms discussed are intended to be used in a special class of problems in which a batch of measurements is only necessary at the initial stage of the solution. At the same time, it should be noted that it is the accumulation of data that makes the algorithm efficient. The relevant explanations are given in the text, lines 541-545.

Reviewer 3 Report

Comments and Suggestions for Authors

  1. For abstract, while author need to briefly explain why algorithm is propose, but author need to focus on briefly explain the algorithm, and what are the advantages of these proposed algorithms
  2. In Page 2, line 48, recursive and sequential could be two different problem. Kalman filter is a sequential algorithm. But optimization algorithm is recursive.
  3. In page 2, could author define single-extermal?
  4. In pasge 3 line 119, for the phrase "which allows identifying it". This phrase is uncleared.
  5. For section 1, it will be great that if author could briefly mention the algorithm is typically designed for which type of application or scenario.
  6. For vector, it is recommended to  bold so that we could differentiate scalar and vector. Please make sure there is no encoding problem, such as in Equation (4), (6) and etc.
  7. For matrix, please bold the capital letter too.
  8. For condition PDF, it is recommend to express in conventional way, such as p(X | Y) instead of p(X / Y). To avoid confusion with division or inverse matrix.
  9. What is "overline{1.l} in page 5 line 193 represent? 
  10. For Page 9, are Fs and Gs in T matrix are time-variance or time-invariant matrices?
  11. It is suggested to shortern section 2 and 3 if possible, if those are not the algorithm proposed by author. As it could easily confuse reader since they took 50% of content pages, excluding simulation, discussion and references.
  12. For figure 5, 6, 11, what are the axes label or represent?
  13. The conclusion section is way too long. It shall briefly summarize the proposed algorithms, how they solve the problem, and how they perform instead.

Author Response

Dear Reviewer,

Thank you very much for giving us the opportunity to revise our manuscript. We sincerely appreciate the time and effort that you have dedicated to providing insightful feedback on our work. We have carefully considered all the comments and found them highly constructive and helpful for us to improve the quality of our paper.

All changes in the text are highlighted in yellow.

Comment 1: For abstract, while author need to briefly explain why algorithm is propose, but author need to focus on briefly explain the algorithm, and what are the advantages of these proposed algorithms

Response 1: We sincerely thank the reviewer for this insightful comment. We have revised the abstract to include a brief description of our algorithm and its advantages. The revised text of the abstract is given below.

A class of nonlinear filtering problems connected with data fusion from various navigation sensors and a navigation system is considered. A special feature of these problems is that the posterior probability density function (PDF) of the state vector being estimated changes its character from multi-extremal to single-extremal as measurements accumulate. The algorithms based on sequential Monte Carlo methods, which, in principle, provide the possibility of attaining potential accuracy, are computationally complicated, especially when implemented in real time. Traditional recursive algorithms, such as the Extended Kalman Filter and its iterative modification prove to be inoperable in this case. Two algorithms, devoid of the above drawbacks, are proposed to solve this class of nonlinear filtering problems. The first algorithm, a Recursive Iterative Batch Linearized Smoother (RI-BLS), is essentially a nonrecursive iterative algorithm; at each iteration, it processes all measurements accumulated by the current time of measurement. However, to do this, it uses a recursive procedure: first, the measurements are processed from the first to the current one in the linearized Kalman filter, and then the obtained estimates are processed recursively in reverse time. The second algorithm, a Recursive Iterative Batch Multiple Linearized Smoother (RI-BMLS), is based on the simultaneous use of a set of RI-BLS running in parallel. The application of the proposed algorithms and their advantages are illustrated by a methodological example and solution of the map-aided navigation problem. Calculation of the computational complexity factor has shown that the RI-BLS is more than 15-fold simpler than the particle filter in computational terms, and the RI-BMLS, more than 20-fold with comparable estimation accuracy.

Comment 2: In Page 2, line 48, recursive and sequential could be two different problem. Kalman filter is a sequential algorithm. But optimization algorithm is recursive.

Response 2: We sincerely thank the reviewer for pointing out this issue. We have removed the mention of a sequential algorithm, since, as you correctly noted, this study deals with a recursive algorithm. Thus, now the text reads as follows (lines 48-49):

Usually, algorithms are designed as recursive ones, which implies processing of incoming measurements one after another.

Comment 3. In page 2, could author define single-extremal?

Response 3: We have followed your advice and added the definition of single-extremal density: it is a probability density function (PDF) that has only one extremum over its entire domain. Hence, now the text reads as follows (lines 49-51):

The problems in which the posterior PDF has a single-extremal form (the PDF has only one extremum over its entire domain, and from here on, such PDF is referred to as single-extremal) are often solved using recursive Kalman-type algorithms (KTA) based on the Gaussian approximation of the posterior density [6,11,13–16].

Comment 4: In page 3 line 119, for the phrase "which allows identifying it". This phrase is uncleared.

Response 4: We thank the reviewer for this technical comment. We have, accordingly, revised the manuscript, so that now the text reads as follows (lines 120-122):

At the moment when the posterior PDF becomes single-extremal, the estimates at the output of the RI-BLS set with various linearization points are grouped into an area corresponding to one extremum, which allows identifying the moment at which the posterior PDF becomes single-extremal.

Comment 5: For section 1, it will be great that if author could briefly mention the algorithm is typically designed for which type of application or scenario.

Response 5: Thank you for the advice. We have accordingly revised the manuscript to address these critical issues. Now the text reads as follows (lines 124-127):

Such algorithms are used to solve the problem of navigation system correction with the use of nonlinear measurements, for example, in navigation of a group of autonomous underwater vehicles, single-beacon navigation, and map-aided navigation.

Comment 6: For vector, it is recommended to bold so that we could differentiate scalar and vector.  Please make sure there is no encoding problem, such as in Equation (4), (6) and etc.

Response 6: We thank the reviewer for this advice. We have introduced bold font for composite vectors. However, it should be explained that bold font is used in the text to denote only composite vectors and block matrices. In our opinion, using bold font for all matrices will make it more difficult to understand what exactly we mean.

Comment 7: For matrix, please bold the capital letter too.

Response 7: Please see Response 6.

Comment 8: For condition PDF, it is recommend to express in conventional way, such as p(X | Y) instead of p(X / Y). To avoid confusion with division or inverse matrix.

Response 8: We thank the reviewer for this technical comment. We have, accordingly, revised the text. We replaced p(xk/Yk) with p(xk|Yk).

Comment 9: What is "overline{1.l} in page 5 line 193 represent? 

Response 9: Thank you for this comment. We have replaced overline{1.l} with a more comprehensible notation 1..k.

Comment 10: For Page 9, are Fs and Gs in T matrix are time-variance or time-invariant matrices?

Response 10: Thank you for this critical comment. We have taken the comment into consideration and made changes to the formulas for these matrices (see Appendix A.4).

Comment 11: It is suggested to shorten section 2 and 3 if possible, if those are not the algorithm proposed by author. As it could easily confuse reader since they took 50% of content pages, excluding simulation, discussion and references.

Response 11: Thank you for the significant advice. We have, accordingly, revised the manuscript. The descriptions of optimal recursive, nonrecursive, suboptimal recursive and nonrecursive (batch) algorithms based on linearization have moved to Appendix A.

Comment 12: For figure 5, 6, 11, what are the axes label or represent?

Response 12: We thank the reviewer for this profound technical comment. We have made corrections in the axes labels in Figures 5, 6, 11.

Comments 13: The conclusion section is way too long. It shall briefly summarize the proposed algorithms, how they solve the problem, and how they perform instead.

Response 13: Thank you for the advice. The text of the conclusion has been revised as follows:

A class of nonlinear filtering problems connected with data fusion from various navigation sensors and a navigation system has been considered. A special feature of these problems is that the posterior PDF of the state vector being estimated changes its character from multi-extremal to single-extremal as measurements accumulate.

Algorithms based on sequential Monte Carlo methods, which, in principle, provide the possibility of attaining potential accuracy, corresponding to the optimal accuracy in the mean square sense of the estimate are computationally complicated, especially when implemented in real time. Traditional recursive algorithms, such as the Extended Kalman Filter and its iterative modification prove to be inoperable in this case.

Two algorithms, devoid of the above drawbacks, are proposed to solve this class of nonlinear filtering problems: the RI-BLS and the RI-BMLS.

The first algorithm, RI-BLS, is essentially a nonrecursive iterative algorithm; at each iteration, it processes all measurements accumulated by the current time of measurement. However, to do this, it uses a recursive procedure: first, the measurements are processed from the first to the current one in the linearized Kalman filter, and then the obtained estimates are processed recursively in reverse time. The RI-BLS, like the I-BLS, provides accuracy close to the potential one and has the consistency property starting from the moment of time , when the posterior PDF becomes single-extremal. Along with this, when implementing the RI-BLS, it is not necessary to invert high-dimensional matrices nk × nk, but it is sufficient to invert only n-dimensional matrices, which allows us to significantly reduce the amount of calculations compared to the I-BLS. This algorithm can be used in problems for which the moment when the posterior PDF becomes single-extremal can be determined in advance. In practice, this is not always the case.

The second algorithm is free from this drawback. It is the RI-BMLS based on the RI-BLS that makes it possible to determine the moment . The essence of the RI-BMLS lies in the simultaneous use of a set of RI-BLS running in parallel; the linearization points for each RI-BLS are selected based on the peculiarities of the problem being solved. The estimates generated by a set of RI-BLS are grouped in the domains corresponding to the PDF extrema, both local and global. At the moment of time , when the posterior PDF becomes single-extremal, the estimates at the RI-BLS output are clustered within a small domain corresponding to a single extremum. This allows identifying , after which the problem is solved using one IEKF, which, in the case of a single-extremal PDF, is consistent and provides accuracy close to the potential one, corresponding to the optimal algorithm.

The application of the proposed algorithms has been illustrated by a methodological example and the solution of the map-aided navigation problem. The effectiveness of the proposed algorithms is demonstrated by solving the problem of data fusion from a navigation system and a sensor of the Earth gravity field, the digital map of which is available on board the vessel. Calculation of the computational complexity factor has shown that the RI-BLS is more than 15-fold simpler than the particle filter in computational terms, and the RI-BMLS, more than 20-fold with comparable estimation accuracy.

Once again, we are truly grateful for your constructive comments. We hope that the revisions have addressed your concerns satisfactorily, and we look forward to your positive decision, remaining fully committed to further improving the manuscript based on your guidance.

Round 2

Reviewer 2 Report

Comments and Suggestions for Authors

The authors have done a good job and revised all of my concerns. I think it suitable to publish. I congratulate the authors on achieving innovative results.

Author Response

We are truly appreciative of your positive assessment of our manuscript. Your recognition of the value and relevance of our research is both humbling and motivating. We are confident that the revisions we have made, guided by your expert feedback, will make our paper even stronger.  We are deeply grateful for your contribution to the improvement of our work.

Reviewer 3 Report

Comments and Suggestions for Authors

  1. Author need to follow the conventional notation, so that it is easier for general reader to understand, without any confusion.
  2. For Page 5 line 186, it should be "1, 2, ... , k" (or 1, ... , k might be acceptable) instead of "1 ... k". Comma shall be indicated. 
  3. Author might want to double check the similar notation as above for the rest. For example line 236 in page 6, it was 0"x0^T, .... xk^T" instead. 
  4. It is strongly for author to bold vector and Matrix for obvious reason. For example, yk^NS in line 404 is a vector. Is yk in (11) a vector? Because in (15), the expression is not valid for vector. Similarly, can be applied for xk in (13), then in (15).
  5. For section 5 (or also in section 4), author need to clearly indicated how earth gravity field is involved in the process. Currently, there isn't any clear indication.
  6. Please provide legend in Figure 10.

Author Response

Dear Reviewer,

We are sincerely grateful for your attentive review of our manuscript. We appreciate the time and effort you have spent providing constructive feedback on our work.

All changes in the text are highlighted in yellow.

Comment 1: Author need to follow the conventional notation, so that it is easier for general reader to understand, without any confusion.

Response 1: We appreciate your comments. We do our best to use conventional notation in the manuscript.

Comment 2: For Page 5 line 186, it should be "1, 2, ... , k" (or 1, ... , k might be acceptable) instead of "1 ... k". Comma shall be indicated.

Response 2: We thank the reviewer for such an attentive reading of the manuscript. The Relevant changes have been made to the text in the line 186 .

Comment 3: Author might want to double check the similar notation as above for the rest. For example line 236 in page 6, it was 0"x0^T, .... xk^T" instead.

Response 3: Thank you for your constructive comment. All necessary commas have been added to the formulas.

Comment 4: It is strongly for author to bold vector and Matrix for obvious reason. For example, yk^NS in line 404 is a vector. Is yk in (11) a vector? Because in (15), the expression is not valid for vector. Similarly, can be applied for xk in (13), then in (15).

Response 4: We thank the reviewer for such a thorough reading of our manuscript. We have corrected all matrix and vector notations throughout the text. We hope that the formulas in our manuscript are now clearer.

Comment 5: For section 5 (or also in section 4), author need to clearly indicated how earth gravity field is involved in the process. Currently, there isn't any clear indication.

Response 5: Thanks again for the insightful comment. We have added the necessary comments clarifying the relationship between the function   and the gravity field, clarifying them in the sentences on lines 422 and 423 and in the legend for Fig. 6.

Comment 6: Please provide legend in Figure 10.

Response 6: Thank you for your constructive comment. We have added a legend in Fig.10

We hope that the new revisions to the manuscript have made our work clearer and more understandable, and we look forward to your positive decision.